# Acute Oncologic Complications: Clinical–Therapeutic Management in Critical Care and Emergency Departments

Nicole Gri [1], Yaroslava Longhitano [2], Christian Zanza [3], Valentina Monticone [4], Damiano Fuschi [5], Andrea Piccioni [6], Abdelouahab Bellou [7], Ciro Esposito [8], Iride Francesca Ceresa [9] and Gabriele Savioli [10,11,*]

[1] Niguarda Cancer Center, ASST Grande Ospedale Metropolitano Niguarda, Piazza dell'Ospedale Maggiore, 3, 20162 Milano, Italy
[2] Department of Anesthesiology and Perioperative Medicine, University of Pittsburgh, Pittsburgh, PA 15260, USA
[3] Italian Society of Prehospital Emergency Medicine (SIS 118), 74121 Taranto, Italy
[4] Department of Otorhinolaryngology, University of Turin, San Luigi Gonzaga Hospital, 10043 Orbassano, Italy
[5] Department of Italian and Supranational Public Law, School of Law, University of Milan, 20122 Milan, Italy
[6] Department of Emergency Medicine, Polyclinic Agostino Gemelli/IRCCS, Catholic University of the Sacred Heart, 00168 Rome, Italy
[7] Department of Emergency Medicine, Institute of Sciences in Emergency Medicine, Guangdong Provincial People's Hospital, Guangdong Academy of Medical Sciences, Guangzhou 510080, China
[8] Unit of Nephrology and Dialysis, ICS Maugeri, University of Pavia, 27100 Pavia, Italy
[9] Department of Emergency Medicine—IRCCS Humanitas Rozzano, 20089 Rozzano, Italy
[10] Emergency Department, IRCCS Fondazione Policlinico San Matteo, 27100 Pavia, Italy
[11] PhD School in Experimental Medicine, Department of Clinical-Surgical, Diagnostic and Pediatric Sciences, University of Pavia, 27100 Pavia, Italy
* Correspondence: gabrielesavioli@gmail.com; Tel.: +39-3409070001

**Abstract: Introduction.** It is now known that cancer is a major public health problem; on the other hand, it is less known, or rather, often underestimated, that a significant percentage of cancer patients will experience a cancer-related emergency. These conditions, depending on the severity, may require treatment in intensive care or in the emergency departments. In addition, it is not uncommon for a tumor pathology to manifest itself directly, in the first instance, with a related emergency. The emergency unit proves to be a fundamental and central unit in the management of cancer patients. Many cancer cases are diagnosed in the first instance as a result of symptoms that lead the patient's admittance into the emergency room. **Materials and Methods.** This narrative review aims to analyze the impact of acute oncological cases in the emergency setting and the role of the emergency physician in their management. A search was conducted over the period January 1981–April 2023 using the main scientific platforms, including PubMed, Scopus, Medline, Embase and Google scholar, and 156 papers were analyzed. **Results.** To probe into the main oncological emergencies and their management in increasingly overcrowded emergency departments, we analyzed the following acute pathologies: neurological emergencies, metabolic and endocrinological emergencies, vascular emergencies, malignant effusions, neutropenic fever and anemia. **Discussion/Conclusions.** Our analysis found that a redefinition of the emergency department connected with the treatment of oncology patients is necessary, considering not only the treatment of the oncological disease in the strict sense, but also the comorbidities, the oncological emergencies and the palliative care setting. The need to redesign an emergency department that is able to manage acute oncological cases and end of life appears clear, especially when this turns out to be related to severe effects that cannot be managed at home with integrated home care. In conclusion, a redefinition of the paradigm appears mandatory, such as the integration between the various specialists belonging to oncological medicine and the emergency department. Therefore, our work aims to provide what can be a handbook to detect, diagnose and treat oncological emergencies, hoping for patient management in a multidisciplinary perspective, which could also lead to the regular presence of an oncologist in the emergency room.

**Keywords:** oncology; oncological emergencies; emergency medicine; crowding; emergency medicine; hematology/medical oncology; internal medicine; palliative care

## 1. Introduction

Cancer, together with cardiovascular diseases, is one of the main causes of mortality in all countries. According to GLOBOCAN 2020 data, 19.3 million people worldwide have been diagnosed with cancer (precisely 18.1 million, if non-melanoma skin cancer is excluded). Furthermore, in 2020, cancer was the main cause of death for 10.0 million people (precisely, 9.9 million if non-melanoma skin cancer is excluded). Surprisingly, female breast cancer is the most frequently diagnosed cancer, with a total of 2.3 million new cases (representing 11.7% of all cancer diagnoses), followed by lung (11.4%), colorectal (10.0%), prostate (7.3%) and stomach (5.6%) cancers. The leading cause of death is lung cancer (1.8 million deaths), followed by gastrointestinal tumors: colorectal (9.4%), liver (8.3%) and stomach (7.7%) cancers. Female breast cancer caused 6.9% of deaths [1–6].

According to the Italian data, 377,000 diagnoses of malignant tumors were estimated in 2020; 52% in the male population and 48% in the female population. In males, the most frequent is prostate cancer (18%), followed by lung (14%), colorectal (12%), bladder (11%), kidney and urinary tract cancer (5%). In females, the most common is breast cancer (30%), followed by colorectal (11%), lung (7%) and thyroid cancer (5%) [7,8].

With regard to mortality, in Italy the ISTAT 2017 data indicate 173,000 deaths attributable to malignant tumors, 95,000 among men and 76,000 among women. The leading causes of cancer death in men are lung, colorectal and prostate cancer; in females, it is breast cancer, followed by lung and colorectal cancer. Overall, 5-year survival is 54% in men and 63% in women. This difference is linked to the fact that, in women, the most frequent cancer is that of the breast, characterized by a good prognosis and survival, also determined by the positive impact of the screening campaign [9].

**Overcrowding and cancer patients.** Overcrowding in emergency departments is an obstacle in the treatment of cancer patients. Overcrowding refers to the imbalance between the need to provide emergency care and the hospital's willingness to provide the service, representing a problem for the entire hospital and not just the emergency department. Therefore, overcrowding further limits the functioning of an emergency department, in association with all the other factors which hinder hospitalization. Overcrowding in the emergency room is an important issue to consider since often in such situations, patients are not only incorrectly treated, but also not even correctly diagnosed. As diagnosis favoring other clinical settings of a non-oncological prerogative are considered, this results in a bad prognosis of the oncological patient. This is a major problem, particularly in small hospital centers, where the support of oncology or a reference oncologist is often lacking, which was already prevalent before the COVID-19 epidemic. Therefore, designing dedicated care pathways within the emergency room could be proposed both for the diagnosis of first-detection neoplasms, and the management of oncological emergencies and palliation of symptoms for the cancer patient.

*Supportive and palliative care: a necessary definition.* In order to understand how important the collaboration between oncology and palliative care is, even in an attempt to integrate them with emergency medicine, it seems necessary to provide the correct definitions of supportive care, palliative care and hospice care. Following these definitions, it becomes clear that supportive care includes palliative care which, in turn, includes hospice [10,11].

Supportive care represents the set of necessary services promoted to those affected by cancer, in order to meet their needs from different points of view (spiritual, informational, emotional, social and physical) during the diagnostic treatment or follow up.

Supportive care also includes the management of cancer-related complications and cancer treatment-related toxicities, wound care, and providing adequate psychological

support. These aspects encompass promoting various elements of prevention and health promotion, as well as survival, palliation and bereavement [11].

Palliative care is defined as supportive care for patients with advanced-stage cancer. It involves all intervention programs used in both hospital and community settings. Ultimately, hospice represents a form of palliative care that supports patients at the end of life, promoting interventions aimed at relieving pain, suffering and the symptoms associated with terminal-stage disease [11–13].

It is now clear that the early referral of patients to palliative care is of primary importance to allow for the global management of cancer patients, not only concerning the relief from suffering but also in establishing a relationship with the patients and the families, and optimally planning the therapeutic process. This could positively influence the prevention of deaths both in hospital settings and in intensive care units [14,15].

This is also reaffirmed by the World Health Organization, which advocates the support of palliative care with other oncological treatments (mainly radiotherapy and chemotherapy), right from the early stages of the disease [14,16].

However, in most cases, the intervention of palliative care takes place in an advanced stage of the disease. Numerous studies conducted in different countries suggest that the postponement would be linked to a purely semantic issue. The term "palliative care" would act as a deterrent, prompting many oncologists to postpone treatment. Ultimately, these studies have shown that the term "palliative care" would evoke more negative images than the term "supportive care," as it is associated with hospice and end-of-life care, reducing the hope of both patients and families [14,17–19]. On the other hand, the clear role of palliative care in improving the results of patients and their caregivers emerges from data in the literature from the last ten years, suggesting the need to integrate oncology and palliative care [17–23].

Furthermore, an increasing number of studies suggest the positive effect of palliative care on many clinical outcomes, not only in terms of quality of life but also in terms of survival [24,25].

The quality of life and the quality of care in the end-of-life setting are other essential points. In fact, several indicators for assessing the quality of end-of-life care have been incorporated into the American Society of Clinical Oncology (ASCO) and the National Quality Forum (NQF) [26,27]. Among these, the administration of chemotherapy or cancer treatment very near death, repeated visits to the emergency room, hospitalization and death in the hospital or in intensive care units (ICU), and delays in patient admission to hospice are considered indicators of poor quality of care [28–30].

## 2. Materials and Methods

For the timeline from January 1981 to April 2023, a search was conducted on the principal scientific platforms: Pubmed, Scopus, Medline, Embase and Google scholar.

Using MeSH database, initially we found a total of 1447 articles, matching "oncology, oncological emergencies, emergency medicine, crowding, emergency medicine; hematology/medical oncology; internal medicine; palliative care". A second screening reduced the number to 440 suitable papers overall, from which were then excluded meeting abstracts, books, manuscripts unavailable, original papers without abstracts and brief reports, and keeping only relevant articles related to oncology and emergency medicine, critical or intensive care medicine and acute medicine.

Moreover, reference lists of each article were reviewed for finding relevant articles to add and, finally, 156 papers were analyzed in this clinical review.

## 3. Discussion

Oncological emergencies represent critical conditions for the patient's life, determined by a disease or the toxicity related to the antineoplastic treatment, and can manifest in various ways. Additionally, infections and hematological conditions can further complicate the clinical picture. When the cause is attributable to the oncological disease, the main treat-

ment consists of specific therapy, along with symptomatic treatment. The most important oncology emergencies are:

- Neurological emergencies (cord compression, intracranial hypertension);
- Metabolic and endocrinological emergencies (tumor lysis syndrome, hyponatraemia and SIAD, hypercalcaemia, hypomagnesaemia, adrenal insufficiency);
- Vascular emergencies (superior vena cava syndrome);
- Neutropenic Fever (NF) [1,31–34].

In the following paragraphs we offer a brief description of the manifestations and treatment of the main oncological emergencies.

### 3.1. Neurological Emergencies

*Spinal cord compression (scc) and intracranial hypertension (*Table 1*).*

**Table 1.** Presentation and management of neurological emergencies: spinal cord compression (scc) and intracranial hypertension.

| Syndrome | Presentation | Management |
|---|---|---|
| **Spinal cord compression** | ❖ A gradual onset may only be demonstrated by back pain and weakness. ❖ **Symptoms:** acute sensory loss, urinary retention, constipation, perineal numbness, sudden inability to walk. | ❖ Neurologic examination ❖ Total spine MRI: gold standard. For patients unable to tolerate MRI, a CT scan with and without IV contrast can be attempted, but the diagnostic yield, because of inferior imaging resolution, is limited. ❖ Opiate pain control and dexamethasone 10 mg IV loading dose followed by 4 mg every 6 h. ❖ Urgent neurosurgical and radiation oncology consultation. ❖ If surgery is not indicated, palliative stereotactic body radiotherapy of 16–24 Gy in 1 fraction or 24–30 Gy in 3 fractions is administered. |
| **Intracranial hypertension** | ❖ **Symptoms:** intractable headaches or nausea and vomiting are most common; (vomiting is correlated with infratentorial lesions and risk brainstem impingement); altered mental status; focal neurologic deficits; vision changes. | ❖ An expedited non-contrast CT scan to evaluate for midline shift is then followed by a non-urgent brain MRI with and without contrast to further characterize the posterior fossa and delineate smaller lesions. ❖ Dexamethasone is recommended except when lymphoma is suspected because this can compromise the diagnostic biopsy by shrinking the mass. ❖ Non-enzyme-inducing anticonvulsants (e.g., levetiracetam) are typically administered in oncology patients because the risk of drug–drug interactions is minimized, although there are mixed data supporting their use. ❖ Hypertonic (3%) saline or mannitol can be used emergently to create an osmotic gradient across the blood–brain barrier; the priority is alleviating the mass effect. ❖ Gamma Knife stereotactic radiosurgery has emerged as the preferred modality for managing solid intracranial masses. ❖ Lumbar puncture is contraindicated because it increases the risk of herniation. |

Spinal cord compression represents an emergency diagnosed in 5–10% of cancer patients, especially in the presence of neoplasms with a marked tendency to bone metastasis (lung, breast and prostate). Only rarely is it related to the presence of a primary tumor affecting the central nervous system.

Spinal cord compression is attributed to three mechanisms:

- Most commonly, due to the invasion of the epidural space by an extradural extramedullary tumor (usually a vertebral body metastasis) or by an intradural extramedullary tumor (due to the presence of a primary central nervous system tumor). This mechanism causes permanent damage to the spinal cord (resulting in paraplegia or tetraplegia) due to the ischemia generated by compression of the venous plexus, leading to intramedullary edema and reduced capillary blood flow due to increased pressure on small arterioles.
- Intramedullary metastases.
- Invasion of the vertebral foramina by paraspinal masses.

Back pain, with characteristic pain and variable localization depending on the level of compression, often resistant to opioids, represents the main onset symptom (in 90–95% of cases). Pain can be associated with sensory neurological deficits (paresthesia, hypoesthesia and anesthesia), motor deficits (weakness up to paralysis) and autonomic nervous system deficits (urinary retention, constipation, incontinence) [35,36].

Regarding the diagnosis, MRI represents the diagnostic gold standard; however, it is not always executable in an emergency regimen. Alternatively, a CT scan of the spine, which provides an adequate representation of bony structures, may be considered [37,38]. From a treatment point of view, corticosteroids should be administered as early as possible to reduce edema. The treatment of choice is frequently radiotherapy (30 Gy in 10 fractions). On the other hand, in selected patients and those with good performance status, surgical treatment (laminectomy or resection and replacement of the vertebral body) is possible.

The main goals of treatment are the preservation of neurological function and the improvement in the quality of life [35,36,39].

The other major neurological oncological emergency is represented by intracranial hypertension, with subsequent edema and cerebral malfunction, up to herniation in the most serious cases. The main cause of intracranial hypertension is the presence and rapid expansion of primary or metastatic intracranial tumors. The main symptoms include headache, nausea, vomiting, visual disturbances, papillary edema, confused states, alterations in the state of consciousness and progressive neurological deficits (gait alterations, paralysis) [38,40].

Imaging techniques represent the diagnostic gold standard (CT, MRI). Treatment should take into account the patient's clinical condition and the prognosis of the underlying disease. Medical therapy is based on the use of glucocorticoids and osmotic diuretics. Specific medical antineoplastic treatments (chemotherapy, targeted therapy) and radiotherapy can effectively contribute to symptom control and disease management. In some cases, neurosurgical treatment may be considered. However, the presence of intracranial hypertension markedly worsens the prognosis, with a median survival of 3 months and a one-year survival of 10% [35,37,41,42].

*3.2. Metabolic and Endocrinological Emergencies (Table 2)*

*Tumor lysis syndrome (TLS).*

Tumor lysis syndrome (TLS) represents an oncological emergency that, if not recognized and treated promptly, is associated with a high risk of mortality (from 29% to 79%). This syndrome is due to the massive destruction of tumor cells, resulting in the release of their contents into the bloodstream [43–45].

The consequent release of cellular metabolites, including potassium, phosphates and uric acid, saturates the renal excretion mechanisms, leading to hyperkalemia, hyperphosphatemia, hyperuricemia and secondary hypocalcemia, which can be fatal. These substances can also precipitate directly inside the renal tubules, resulting in acute renal failure [46,47].

The risk is higher in the presence of aggressive tumors with high chemosensitivity, including both hematological malignancies (acute leukemias and high-grade non-Hodgkin's lymphomas) and solid malignancies (SCLC and testicular tumors) [48,49].

Patients who develop this clinical condition must be promptly treated by correcting the metabolic alterations that have arisen. Additionally, careful monitoring of cardiac activity, diuresis, plasma concentration of electrolytes and uric acid, LDH, and creatinine is required. Despite prophylactic measures and treatments to reduce acute kidney injury, a small proportion of patients (approximately 5%) may require hemodialysis [50,51].

Hyponatremia and SIAD. Hyponatremia is defined as a serum sodium concentration below 135 mmol/L [52].

Hyponatremia is classified, according to the speed of onset, as acute or chronic (when onset is less or more than 48 h, respectively). The symptomatology is predominantly neurological, and the clinical manifestations depend on the severity and speed of onset. In the presence of mild or chronic hyponatremia, patients are often asymptomatic or have nonspecific symptoms (asthenia, nausea, headache). Symptoms such as headache, lethargy, convulsions, and coma may occur in severe or acute hyponatremia. Additionally, hyponatremia can be classified according to plasma osmolarity (hyposmolar or hyperosmolar) and volume (euvolemic, hypervolemic, hypovolemic) [53–55].

Decreased plasma sodium concentration is the most frequent electrolyte disturbance in cancer patients, especially in patients with lung, prostate, pancreatic and renal cancer. Furthermore, hyponatremia represents a negative prognostic factor for patients affected by cancer and, therefore, must be corrected.

However, in most cancer patients, hyponatremia is a consequence of the syndrome of inappropriate antidiuresis (SIADH), which should be suspected in the presence of hyposmolar and euvolemic hyponatremia. SIADH represents, in most cases, a paraneoplastic syndrome due to an ectopic secretion of vasopressin by the tumor, which leads to excessive water retention and a reduction in plasma sodium concentration. SIADH has an incidence of 1–2% in the oncological population, reaching 15% in patients with SCLC. Although it may also be associated with breast cancer, lymphoma and neck tumors [56–62].

It should be noted that some chemotherapy drugs or pain relievers, including opioids, can also cause SIADH [63–65].

The treatment is based on the antineoplastic therapy of the underlying disease, and other measures must be taken in relation to the clinical condition. In patients with few symptoms, the use of AVP-receptor antagonists can be considered, while in patients with severe symptoms, a hypertonic solution remains the treatment of choice [66–68].

Hypercalcemia. Hypercalcemia is the most common metabolic cancer emergency, with a frequency of 10–30% among the cancer population. Hypercalcemia is defined as a corrected serum calcium concentration > 10 mg/dL and/or an ionized calcium concentration > 5.6 mg/dL.

This electrolyte disorder is most frequently associated with breast, lung, kidney, thyroid and head and neck cancer. Regarding hematological cancers, it is more frequent in lymphomas. It is also more common in the advanced disease setting and is associated with a poor prognosis [33,69–71].

Four mechanisms underlying neoplastic hypercalcemia have been recognized:

- Production of a PTH-like protein (PTHrp: parathyroid hormone-related protein);
- Ectopic production of PTH;
- Osteolytic metastases;
- Release of 1,25-dihydroxycholecalciferol (especially in lymphomas) [72–74].

The symptomatology varies based on the speed of onset and the severity of the alteration, similar to what happens in hyponatremia. Symptoms can be non-specific, affecting various organs and systems, with neurological disorders (fatigue, muscle weakness, reduced reflexes, apathy, lethargy, behavioral changes up to coma), gastrointestinal disorders (loss of appetite, nausea, vomiting, constipation, paralytic ileus), cardiovascular issues (bradyarrhythmias, ECG changes) and urinary disorders (polyuria, polydipsia, acute kidney injury, nephrolithiasis) [72,75].

Similar to the clinic, the therapy for hypercalcemia also depends on the serum calcium concentration and the severity of the symptoms. The first step to take is the administration of intravenous fluids, often having to resort to hemodialysis in patients with acute kidney injury.

In the presence of severe hypercalcemia unresponsive to hydration, the use of bisphosphonates should be considered. In fact, bisphosphonates cause a reduction in serum calcium starting from 12 to 48 h after administration, with a persistent effect for 2–4 weeks. Instead, treatment with glucocorticoids appears more effective in patients with calcitriol-producing tumors and lymphomas, both Hodgkin's and non-Hodgkin's [71,76–79].

**Table 2.** Presentation and management of metabolic emergencies.

| Syndrome | Presentation | Management |
|---|---|---|
| **Syndrome of inappropriate antidiuretic hormone (SIADH)** | ❖ Urine osmolarity > 100 mOsm (coincident with euvolemic hypotonic hyponatremia)<br>❖ Corresponding serum Na levels:<br>  - Mild: 130–134 mEq/dL<br>  - Moderate: 125–129 mEq/dL<br>  - Severe: <125 mEq/dL<br>❖ Acute hyponatremia with headaches or neurocognitive slowing.<br>❖ Severe hyponatremia can be associated with seizures or death.<br>**REMEMBER:**<br>❖ Must be distinguished from hypovolemic hyponatremia (urine osmolarity > 300 mOsm; urine sodium < 20 mEq/L). Main cause: excessive gastrointestinal loss. | ❖ *Symptomatic hyponatremia*: 100 mL 3% normal saline bolus to acutely raise serum sodium by 2–3 mEq/L.<br>❖ *Chronic hyponatremia*: Free water restriction and sodium chloride tablets.<br>**REMEMBER:**<br>❖ Total increase in serum sodium by no more than 4–6 mEq/L in 24 h to avoid central pontine myelinolysis.<br>❖ Correction of hyponatremia is usually necessary prior to initiating systemic therapy. |
| **Hypercalcemia** | ❖ Corresponding serum Ca levels:<br>  - Mild: 10.5–11.9 mg/dL<br>  - Moderate: 12.0–13.9 mg/dL<br>  - Severe: ≥14.0 mg/dL<br>❖ In the case of hypoalbuminemia, observed serum calcium must be further increased by 0.8 * (4.0—serum albumin) mg/dL.<br>❖ Presentation: altered mental status, muscle weakness, constipation, dehydration with ensuing acute kidney injury, urolithiasis (in subacute presentations). | ❖ Immediate aggressive intravenous hydration with normal saline (1–2 L in the first hour, followed by 2 L at 200 mL/hr with close monitoring of volume status).<br>❖ Early intravenous bisphosphonate administration (most commonly zolendronic acid).<br>❖ Supplemental calcitonin can be administered during the first 48 h while the bisphosphonate is not yet at peak efficacy.<br>Denosumab is alternatively administered in bisphosphonate-refractory cases.<br>❖ Loop diuretics are generally now avoided as they can exacerbate hypercalcemia and kidney injury in inadequately hydrated patients. |

*Hypomagnesaemia.* Hypomagnesemia is defined as a serum magnesium concentration < 1.8 mg/dL [80].

Different categories of patients are at risk of developing hypomagnesemia, which can occur in up to 50–60% of cases in hospitalized or critically ill cancer patients [81].

There are several causes that can lead to hypomagnesemia in oncological patients, such as: hypoalbuminemia, chemotherapy based on platinum derivatives; therapy with anti-EGFR drugs (Cetuximab, Panitumumab), which inhibit the magnesium channel regulating renal and gastrointestinal transport; hypercalcemia (due to competition of calcium on the same transporter as magnesium in the loop of Henle); prolonged diarrhea; blood transfusions; drugs that interfere with renal tubular magnesium transport (loop diuretics);

bisphosphonates; corticosteroids; proton pump inhibitors; antibiotics; antiviral drugs; short bowel syndrome; and pancreatic insufficiency (post-surgery) [82–94].

Symptoms of hypomagnesemia include inappetence, nausea, vomiting and asthenia. An emergency condition can be indicated by the onset of behavioral alterations (confusion, agitation, depression), neuromuscular hyperexcitability (cramps, hyperreflexia of deep tendon reflexes, muscle contractions), and arrhythmias or ECG-graphic alterations (lengthening of the QT interval, ventricular tachycardia, torsade de pointes, ventricular fibrillation) [81,82,90].

However, the measurement of magnesemia is not a routine examination in cancer patients and is reserved only for patients receiving Cetuximab and Panitumumab therapy. Therefore, this condition must be detected based on clinical symptoms. In particular, hypomagnesemia should be suspected in the presence of hypocalcemia and hypokalemia, especially in patients taking the previously mentioned drugs.

The treatment is based on magnesium supplementation, depending on the dosage of magnesemia. The treatment for grade 2 hypomagnesemia (<1.2 mg/dL) involves using oral magnesium oxide, magnesium gluconate, or sulfate. For grade 3–4 hypomagnesemia (<0.9 mg/dL), the use of intravenous magnesium sulfate is necessary and cancer treatment should be suspended.

Significant magnesium supplementation is also required in patients with diarrhea, vomiting, or flu syndrome with an initial state of dehydration or major surgery [95–98].

Adrenal insufficiency. Adrenal insufficiency is a life-threatening disorder caused by deficient glucocorticoid production, with or without mineralocorticoid underproduction, by the adrenal gland. In cancer patients, this condition often results from inadequate interruption or suspension of ongoing steroid therapy, leading to altered adrenal gland function or, rarely, from the use of immunotherapy drugs (anti-CTLA4 and anti-PD1). Less frequently, adrenal insufficiency arises from an alteration of the hypothalamic-pituitary axis, due to the altered production of corticotropin and, therefore, cortisol [99,100].

Adrenal insufficiency can occur acutely or chronically. Autoimmune-based hypophysitis can occur in patients treated with anti-CTLA4 drugs alone (mean incidence 13%) or in combination with anti-PD1 drugs and, less frequently, with anti-PD1 or anti-PDL1 drugs alone. Male gender and advanced age are risk factors for the onset of immunotherapy-related hypophysitis.

From a neuroradiological point of view, upon MRI of the brain, the pituitary is often enlarged. This sign, although not always present, is a specific and sensitive indicator of hypophysitis that precedes clinical symptoms and laboratory alterations [101–104].

The timely diagnosis of iatrogenic hypophysitis is based on clinical suspicion and laboratory tests (hormonal dosage of ACTH and cortisolemia), performed at baseline and then at regular intervals. The diagnostic difficulty is often also linked to the presence of non-specific symptoms, often not unequivocally attributable to endocrinopathy. Symptoms of adrenal insufficiency include general malaise, asthenia, nausea, confusion, headache, hypovisus, orthostatic hypotension, abdominal pain, fever and, finally, coma [105–107].

Secondary autoimmune adrenal insufficiency from antineoplastic drugs is usually permanent and requires chronic supplemental treatment. In emergency or urgency settings, high-dose hydrocortisone treatment and hydration with physiological solution are necessary. Hydrocortisone is the preferred steroid for subsequent chronic therapy, with 2–3 administrations per day, simulating the circulating levels of glucocorticoids based on the circadian rhythm of secretion [108,109].

### 3.3. Vascular Emergencies (Table 3)

*Superior vena cava syndrome.*

Superior Vena Cava Syndrome (also known as mediastinal syndrome) occurs as a result of an obstruction to blood flow in the superior vena cava. The thrombotic phenomenon in oncological patients can be due to various causes, including direct invasion of the superior vena cava by the neoplasm, extrinsic compression, or the presence of a central venous catheter.

This syndrome is determined by the presence of malignant tumor pathology in 80% of cases. The tumors most frequently associated with superior vena cava syndrome include lung cancer, Hodgkin's lymphoma and metastases. The cardinal symptoms, which can manifest acutely, constituting an emergency situation, are dyspnea, cough, swelling of the neck, confusion and headache. Other typical signs include cyanosis, plethora, edema and distended veins on the face, neck and chest [110–113].

Diagnosis is based on radiological imaging (chest CT), which identifies not only the cause but also the degree of superior vena cava obstruction and collateral venous supply. Radiotherapy is an emergency treatment. A targeted therapy based on the etiology is then necessary: chemotherapy, radiotherapy and possible placement of a stent in the case of tumor pathology; removal of the central venous catheter or fibrinolysis in the case of non-tumor disease [114–116].

*Venous Thromboembolism.*

Cancer-associated thrombosis includes both venous thromboembolism (VTE) and arterial events. Despite improvements in cancer treatment, the incidence of this event has increased in recent years and is associated with decreased survival and worsening quality of life, including increased mortality. This event would be related mainly to the type of tumor (pancreatic, stomach and nervous system tumors have the highest risk), risk factors related to the patient and the specific oncological treatment. Several mechanisms correlated with the onset of thrombosis have been highlighted: the most important would be determined by the release of procoagulant substances by the tumor itself, with consequent activation of the coagulation cascade and platelets (such as, for example, tissue factor and podoplanin) [117–120].

The suspected diagnosis is mainly based on the clinic, as signs and symptoms may be nonspecific. Pretest probability assessments, laboratory tests and specific diagnostic tests (color Doppler ultrasound, ultrasonography, CT) are used for diagnosis. The therapy is based on the use of anticoagulant drugs (low molecular weight heparin or direct oral anticoagulants), also taking into account the risk of bleeding and any patient co-pathologies. Therefore, individualized treatment should also be considered in the presence of VTE [117,121,122].

### 3.4. Malignant Effusions

Malignant effusions in cancer patients cause a set of significant symptoms, often associated with anguish and a feeling of imminent death, leading to a poor prognosis. The treatment is mainly aimed at managing and relieving symptoms, favoring minimally invasive interventions. Among malignant effusions, the most important are pericardial effusion and malignant pleural effusion [123].

Malignant pericardial effusion is a serious manifestation of malignant tumors. It can be caused by both solid neoplasms and hematological neoplasms, while primary tumors of the pericardium are quite rare. In cases where a neoplasm is the main cause, the effusion can form through direct invasion of the pericardium or as a result of metastasis. Not infrequently, the pericardial effusion may represent the initial manifestation of the neoplasm. In particular, malignancy should be excluded in the presence of acute pericardial disease resulting in tamponade at onset, the presence of a recurring effusion, or a rapidly increasing effusion. Therefore, identifying the cause of the pericardial effusion plays a fundamental role from a therapeutic and prognostic perspective. In emergency settings, due to the presence of symptoms related to massive effusion or cardiac tamponade, emergency pericardiocentesis often results in rapid symptomatic relief. The treatment of cancer patients with malignant pleural effusion is based on multidisciplinary sharing, but an individual treatment plan is undoubtedly mandatory, taking into account various factors related to pericardial effusion itself (hemodynamic impact, size), the tumor (stage of the disease, prognosis) and the patient's performance status and comorbidities [124–129].

**Table 3.** Presentation and management of vascular emergencies: superior vena cava syndrome and venous thromboembolism.

| Syndrome | Presentation | Management |
|---|---|---|
| **Superior Vena Cava Syndrome** | ❖ Facial edema and subcutaneous vein engorgement in head, neck and chest. <br> ❖ Complete obstructions additionally present with plethora, dyspnea, orthopnea, cough, hoarseness, cyanosis, headache, seizures and, eventually, coma. <br> ❖ Chest X-ray: mediastinal widening (66%) or pleural effusions (25%). <br> ❖ CT with contrast: gold standard. | ❖ Elevate head to minimize venous congestion. <br> ❖ Medical and radiation oncology consultation should be expedited to initiate systemic therapy because reducing tumor bulk is definitive. <br> ❖ Urgent thrombolysis, thrombectomy, or placement of a venous stent may alleviate stridor and hemodynamic compromise although vascular intervention risks luminal perforation. <br> ❖ Diuretic use should be minimized. |
| **Venous Thromboembolism** | ❖ Chief complaints of shortness of breath, unilateral leg swelling, or reduced oxygenation on pulse oximetry. <br> ❖ D-dimer levels are not informative: they can be elevated generally in cancer patients. | ❖ CT angiography of the chest is the definitive study because not only can it rule out other processes but it can also confirm right ventricular strain. <br> ❖ If IV contrast is contraindicated, a ventilation-perfusion scan along with cardiac echography is appropriate. <br> ❖ Systemic thrombolysis is indicated for massive PE with hemodynamic compromise except in patients with a high risk of bleeding, for whom catheter-assisted thrombectomy is indicated. <br> ❖ Factor Xa inhibitors are noninferior to low-molecular-weight heparin, with apixaban demonstrating fewer major bleeding events. <br> ❖ The benefit of thromboprophylaxis has not been demonstrated. <br> ❖ Patients with small, incidental PEs and no functional or vital sign compromise are eligible to initiate anticoagulation in the ED then be safely discharged home with close follow-up. |

Malignant pleural effusion is also a common manifestation in patients with neoplastic disease. Potentially, all cancers can produce a pleural effusion, which can be the first manifestation in approximately 10% of patients. Similar to pericardial effusion, the presence of a malignant pleural effusion is linked to advanced disease, poor prognosis and deterioration in the quality of life. Most malignant pleural effusions are related to the presence of pleural metastases, most frequently from lung and breast cancer, or to primary thoracic tumors. The main symptom is dyspnea, associated with alterations in respiratory parameters. Regarding imaging, CT is generally considered the gold standard, although chest ultrasonography is also useful in hypothesizing the extent of the effusion and identifying the presence of diaphragmatic or pleural nodules and suspicious thickenings. Cytology is obtained through pleural fluid aspiration or pleural biopsy. Treatment is mainly palliative, aiming at palliation and relief of symptoms. Interventions include drainage with thoracentesis, indwelling pleural catheter, or pleurodesis. Also in this case, the approach must be based on the amount of the effusion, the characteristics and stage of the oncological disease, the patient's prognosis and their performance status [130–135].

*3.5. Neutropenic Fever (Table 4)*

Neutropenic fever (NF) is one of the best-known and most frequent oncological emergencies. The frequency of febrile neutropenia differs according to the type of malignancy: patients with hematological tumors have an increased risk (equal to 80%) of developing febrile neutropenia during chemotherapy treatment, against 10–50% of patients with solid

tumors. The main risk factors for developing fever include the duration and severity of neutropenia. Febrile neutropenia is defined as a single temperature measurement (oral or axillary) ≥38.3 °C or a temperature ≥ 38°C for 60 min in a patient with an absolute neutrophil count (ANC) < 500/µL [33,136–140].

A precise etiologic cause is difficult to determine in the presence of a single episode of febrile neutropenia. Febrile neutropenia can result from several etiologies, including the underlying cancer (for example, leukemia), toxicity of chemotherapeutic agents, or infection. However, infection is documented in only 20–30% of cases [7,8]. In the presence of an infection, this is more likely supported by bacteria of the endogenous intestinal flora (for example, Escherichia coli, Enterobacter), of the respiratory tract (Streptococcus), or of the skin (Staphylococcus, Streptococcus) [141,142].

In recent decades, there has been a change in the bacterial epidemiology associated with febrile neutropenia. In fact, Gram-positive bacterial infections have increased in incidence. Among the causes that have led to this increase are an increase in the community load of Staphylococcus and an increase in indwelling catheters. However, there has also been an increased incidence of Clostridium difficile-associated infections [137–139,143,144].

In the presence of suspected febrile neutropenia, it is necessary to proceed with a diagnostic evaluation and the possible administration of antibiotics. Initial testing should include a complete blood count, metabolic panel, urinalysis and urine culture, chest X-ray and two sets of blood cultures (including one from an indwelling line, if present). If diarrhea is present, testing for Clostridium difficile and fecal cultures should be considered [140,145].

Broad-spectrum antibiotic therapy should be given within 60 min of identification of febrile neutropenia and after cultures are obtained. The choice of empirical antibiotic should be guided by the susceptibility test and the institution [146–148].

All the pictures previously described can represent the reason for accessing the EDs or be occasional findings on different reasons for accessing. In fact, the acute conditions of cancer patients are often complex to diagnose. Neutropenic fever could, for example, cause syncope with a fall and subsequent head trauma. Conversely, clinical pictures due to electrolyte disturbances or acute renal failure could be the reason for admission to the ED, but diagnostic investigations could highlight pulmonary thromboembolism that occurred without symptoms. All these clinical conditions require intervention with specialized skills that treat these complex patients in full, integrating specific skills on cancer emergencies with skills on other emergencies and an ED management aimed at improving the quality and safety of the facility and the patient [149–157].

*3.6. Anemia*

Anemia is one of the most common laboratory abnormalities and represents one of the most common diagnoses in cancer patients. In this subgroup of patients, the causes of anemia are often multifactorial, related both to direct and indirect effects of the neoplasm, and to the effects of pharmacological treatments (mainly chemotherapy) [158,159].

Anemia is a condition in which the concentration of hemoglobin (Hb) and/or the number of red blood cells are lower than the normal limit, and therefore insufficient to satisfy the physiological needs of an individual, above all in terms of oxygen transport [160,161].

According to the World Health Organization (WHO) criteria and the CTCAE (v5.0) grading system for anemia, the grading of anemia in cancer patients is reported in the Table 5.

Cancer-related anemia is due to three main mechanisms:

- Ineffective erythropoiesis;
- Hemolysis;
- Loss of blood.

These mechanisms can lead to anemia individually or in combination. However, the causes of anemia can best be subclassified into three broad, related categories: production, destruction and loss (bleeding).

**Table 4.** Presentation and management of febrile neutropenia.

| Syndrome | Presentation | Management |
|---|---|---|
| **Superior Vena Cava Syndrome** | ❖ A single oral temperature measured at ≥38.3 °C (101 °F) or a temperature of ≥38.0 °C (100.4 °F) sustained over a 1 h period combined with severe neutropenia, defined as an absolute neutrophil count (ANC) <500 cells/mm$^3$ or an ANC that is expected to decrease to <500 cells/mm$^3$ during the next 48 h. Presentation can bridge the spectrum from otherwise asymptomatic to moribund with florid hemodynamic collapse. | ❖ The current door-to-antibiotics guideline is <1 h after ED presentation, although recent research has shown that delays up to 3 h do not substantially affect outcomes.<br>❖ Blood cultures, both peripheral and from indwelling lines, should be universally collected.<br>❖ Empiric monotherapy with an antipseudomonal β-lactam is recommended with the following exceptions:<br>- Carbapenems for patients with a high risk of infection by an extended-spectrum β-lactamase-expressing organism.<br>- Polymyxin-colistin in patients at high risk for a carbapenemase-producing Klebsiella.<br>- Aztreonam, fluoroquinolones, or aminoglycosides for patients with an anaphylactic β-lactam allergy.<br>❖ Vancomycin, linezolid, or daptomycin should be given only when either methicillin-resistant Staphylococcus aureus or indwelling catheter infections are likely and should be discontinued once these are ruled out.<br>❖ Empiric antifungal coverage is not recommended at the time of presentation. |

**Table 5.** Anemia Grade.

| Anemia Grade | Women [Hb] | Men [Hb] |
|---|---|---|
| Grade 1 | Hb 11.9–10.0 g/dL | Hb 12.9–10.0 g/dL |
| Grade 2 | Hb 9.9–8.0 g/dL | Hb 9.9–8.0 g/dL |
| Grade 3 | Hb ≤ 7.9 g/dL | Hb < 7.9 g/dL |
| Grade 4 | Life-threatening consequences requiring urgent intervention, such as RBC transfusion | Life-threatening consequences requiring urgent intervention, such as RBC transfusion |

These causes can then be related to three categories:

- Drug-induced;
- Induced by infectious diseases;
- Induced by the tumor itself, both solid tumors (especially gastrointestinal tumors) and blood tumors (acute or chronic leukemia);
- Induced by vitamin deficiencies (vitamins B9, B12 [159,162–164]).

Table 6 shows the mechanisms of anemia induced by the different categories of oncological drugs.

Therefore, with this in mind, a systematic approach is needed to identify the causes that lead to anemia. On the other hand, it must be remembered that a first detection of anemia must be carefully investigated, also taking into consideration, among the possible differential diagnoses, the presence of a solid or hematological neoplasm. Furthermore, the correct differential diagnosis of the cause of anemia in patients with cancer is essential to establish the correct treatment and reduce transfusion requirements, with the ultimate aim of improving both quality of life and survival, through improvement in cognition, fatigue and exercise tolerance (Table 7) [158,159,165].

**Table 6.** Iatrogenic anemias from oncological drugs.

| Production (Decreased) | Destruction (Increased) | Loss (Overt, Occult, or Iatrogenic Bleeding) |
|---|---|---|
| Myelosuppressive chemotherapy | Antibiotics (e.g., β-lactams, dapsone) | Anticoagulation (e.g., DOACs, LMWHs, warfarin) |
| Radiation therapy | Chemotherapy (e.g., gemcitabine) | Antiplatelet agents (e.g., clopidogrel, prasugrel, ticagrelor) |
| Tyrosine kinase inhibitors (delayed maturation) | Immunotherapy (e.g., nivolumab, pembrolizumab, ipilimumab) | Nonsteroidal anti-inflammatory drugs |
| Immunotherapy (inflammation) | Intravenous immunoglobulin G | Over-the-counter supplements (e.g., turmeric) |

**Table 7.** Presentation and management of severe anemia.

| Syndrome | Presentation | Management |
|---|---|---|
| **Severe anemia** | ❖ Symptoms: debilitating fatigue and dyspnea, initially with exertion then at rest | ❖ Laboratory exams: reticulocyte count and haptoglobin, iron studies, serum vitamin B12 and folate levels, liver function tests (including albumin, prealbumin, total and direct bilirubin) and lactate dehydrogenase concentrations, as well as a peripheral blood smear, should be obtained in addition to the complete blood count <br> ❖ Hemagglutinins can be assessed to rule out hemolysis <br> ❖ Transfusion should be used as sparingly as possible and should target a hemoglobin level of 7.0 mg/dL unless the patient has established cardiac disease or continues to be overtly symptomatic where 8.0 mg/dL is appropriate <br> ❖ Erythropoiesis-stimulating agents are no longer indicated for cancer-related anemia |

However, blood transfusions in cancer patients, used in the presence of a life-threatening condition but also in anemia of grades other than 4, depending on the needs and clinical status of the patient, have been linked to an increased risk of thrombosis, transmission of pathogens, transfusion reactions, volume and iron overload and, ultimately, also decreased survival. In fact, red blood cell transfusions are associated with a 10-fold greater risk of morbidity than IV iron (1 in 21,413 for RBC vs. ~1 in 200,000 for current IV iron products). Furthermore, it is useful to underline that, although each unit of PRBC contains about 250 mg of iron, this is not immediately bioavailable due to the average life of the transfused red blood cell (90 days). It is, therefore, the clinician's duty to consider both the clinical indicators and the laboratory analyzes to lean towards one or the other treatment, remembering that the transfusion of red blood cells, which requires about 1 or 2 h per unit to be administered, remains an option for the treatment of anemia associated with grade 2 to 4 cancer when other therapies have failed [166–168].

## 4. Conclusions

In order to recognize and treat these severe presentations, careful clinical detection is required, aided by advanced imaging techniques and laboratory images. In this context, the emergency department to which these patients often refer must be able to guarantee early diagnosis and targeted treatment, which significantly impacts the patient's clinical outcome, especially when this emergency is potentially lethal [1,31].

Furthermore, palliative care is an essential component of emergency medicine. In fact, many patients with a terminal illness go to the emergency department for the management of symptoms preceding end of life. In particular, two of the most common and distressing symptoms at end of life are pain and dyspnea, often associated with other symptoms, including nausea and vomiting. From this point of view, the role of the emergency physician

appears crucial, not only in the treatment of symptoms but also in communicating with patients and families [31,149,150].

In the face of the various circumstances for which the cancer patient goes to the emergency department, the need to be able to guarantee practical solutions for the management of clinically different situations appears clear. Such solutions must be methodical and quick in nature but must not be separated from the overall care of the patient, which also includes attention to their family and companions.

The redefinition of the emergency department, therefore, cannot ignore the correct management of the cancer patient, through the establishment of dedicated pathways, even in a multidisciplinary setting, for the targeted treatment of acute cases, as well as the establishment of specific areas dedicated to palliation.

**Author Contributions:** Conceptualization, N.G.; methodology, I.F.C.; software, D.F.; resources, A.P.; data curation, V.M.; writing—original draft preparation, N.G.; writing—review and editing, C.Z. and Y.L.; visualization, A.B.; project administration, C.E.; funding acquisition, C.Z. and G.S. All authors have read and agreed to the published version of the manuscript.

**Funding:** This research received no external funding.

**Conflicts of Interest:** The authors declare no conflict of interest.

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
