# Peer review of "Acute Oncologic Complications: Clinical–Therapeutic Management in Critical Care and Emergency Departments"

_curroncol, doi:10.3390/curroncol30080531_

Round 1
Reviewer 1 Report
Dear authors,
thank you very much for your manuscript on a important topic.
Unfortunately the method of your your review is not described good enough. I recommend a major revision with a detailed description of your method used and the type of review you are presenting. Please present the aim of your work clearly. Please refer also to used guidelines. The abstract should be divided into the usual four parts: Introduction, materials and methods, results and discussion/conclusion.
I would love to see a better structured revision with more in detail descriptions of your work.
Please revise your manuscript.
Author Response
- Unfortunately the method of your your review is not described good enough. I recommend a major revision with a detailed description of your method used and the type of review you are presenting.
Dear reviewer,
Thank you for your clarification and for your request.
Our article is a narrative review, with the aim of analyzing the main oncological emergencies, as well as the evolution of their characterization and treatment through the years need in the emergency medicine setting, in accordance with the most important guidelines and the most recent publications.
Thanks for the suggestion, we thought we'd rewrite materials and methods according to your advice, as follows:
“In order to write a narrative review with the aim of analyzing the main oncological emergencies from the point of view of emergency medicine, also considering the evolution of their understanding and treatment over the years, a search was conducted over the period January 1981 - April 2023 using the main scientific platforms, including Pubmed, Scopus, Medline, Embase and Google scholar. Using MeSH database, we found a total of 1447 articles, matching “oncology, oncological emergencies, emergency medicine, crowding, emergency medicine; hematology/medical oncology; internal medicines; palliative care". Starting from these articles, which covered not only a vast array of topics, but also a very broad time span, a second screening reduced to 440 suitable papers excluding overall: meeting abstracts, books, manuscripts unavailable, original papers without abstracts, brief reports;keeping only relevant article related to oncology and emergency medicine, critical or intensive care medicine and acute medicine. Moreover, references list of each article were reviewed not finding relevant articles to add and finally 156 papers were analyzed in this clinical narrative review. Regarding diagnostic algorithms and treatment, we referred to AIOM, ESMO and NCCN guidelines.”
- Please present the aim of your work clearly.
Dear reviewer,
thank you for your clarification and for your request.
The purpose of our work is to outline, both from a clinical and treatment point of view, the main oncological emergencies and their impact in the field of emergency medicine.
Cancer emergencies represent a health problem that is too little discussed and that should be adequately understood even by the emergency medicine physician. In particular, this represents a fundamentally important topic, due to the increase in cancer diagnoses and therefore the number of patients who can potentially manifest them. Furthermore, the problem of overcrowding in the emergency room also represents a further factor to consider, since often these patients are not only not correctly treated, but not even correctly diagnosed, favoring other clinical settings of a non-oncological prerogative, being the oncological patient considered with a bad prognosis. This represents a problem above all in small hospital centers, where the support of oncology or a reference oncologist is often lacking. Therefore, our work aims to provide, through a review of the literature, what can be a handbook of the main oncological emergencies, to suspect, diagnose and treat them, always hoping for patient management in a multidisciplinary perspective, which could also lead to the permanent figure of an oncologist in the emergency room.
- Please refer also to used guidelines.
Dear reviewer,
thank you for your clarification and for your request.
The guidelines we use include the AIOM, ESMO and NCCN guidelines.
- The abstract should be divided into the usual four parts: Introduction, materials and methods, results and discussion/conclusion.
Dear reviewer,
thank you for your clarification and for your request. Thanks for the suggestion, we thought we'd rewrite the abstract according to your advice, as follows:
Abstract: introduction: It is now known that cancer is a major public health problem; on the other hand, it is less known, or rather, often underestimated, that a significant percentage of cancer patients will experience a cancer-related emergency. These conditions, depending on the severity, may require treatment in intensive care or be treated within the emergency departments. In addition, it is not uncommon for a tumor pathology to manifest itself directly, in the first instance, with a related emergency.
The emergency unit proves to be a fundamental and central unit in the management of the cancer patient. Many cancer cases are diagnosed in the first instance as a result of symptoms that lead the patient to the emergency room.
Materials and methods. this narrative review, has the aim to analize the impact of oncological acute cases in the emergency setting and the role of the emergency physician in their management. A search was conducted over the period January 1981 - April 2023 using the main scientific platforms, including Pubmed, Scopus, Medline, Embase and Google scholar, analyzing 156 papers.
Results. The result of our narrative review was to analyze the main oncological emergencies and their management in increasingly overcrowded emergency departments. The acute pathologies analyzed were: neurological emergencies, metabolic and endocrinological emergencies, vascular emergencies, malignant effusions, neutropenic fever and anemia.
Discussion/conclusions. From this point of view, a redefinition of the emergency department connected with the treatment of the oncological patient is necessary, considering not only the treatment of the oncological disease in the strict sense, but also the comorbidities, the oncological emergencies and the palliative care setting. The need to redesign an emergency department that is able to manage oncological acute cases and the end of life appears clear, especially when this turns out to be related to acute facts that cannot be managed at home with integrated home care. In conclusion, a redefinition of the paradigm appears mandatory, just as the integration between the various specialists belonging to oncological medicine and the emergency department. Therefore, our work aims to provide what can be a handbook of the main oncological emergencies, to suspect, diagnose and treat them, always hoping for patient management in a multidisciplinary perspective, which could also lead to the permanent figure of an oncologist in the emergency room.
Rewriting the abstract, we decided to insert the part relating to overcrowding in the introduction paragraph, considering it one of the problems that afflicts the emergency room, with a consequent impact also on cancer patients. Overcrowding and cancer patients. Overcrowding in emergency departments also represents a problem in the treatment of cancer patients. Overcrowding refers to the imbalance between the need to provide emergency care and the hospital's willingness to provide the service, representing a problem for the entire hospital and not just the emergency room. Therefore, overcrowding represents a further limit to the functioning of the emergency department, in association with all the other factors which hinder hospitalisation. Furthermore, the problem of overcrowding in the emergency room also represents a further factor to consider, since often these patients are not only not correctly treated, but not even correctly diagnosed, favoring other clinical settings of a non-oncological prerogative, being the oncological patient considered with a bad prognosis. This represents a problem above all in small hospital centers, where the support of oncology or a reference oncologist is often lacking. This problem was already relevant before the COVID-19 epidemic, which further had a negative impact in this sense, limiting access to treatment for different categories of patients, including cancer patients. Exactly as happened during the COVID-19 epidemic, the design of dedicated care pathways within the emergency room could also be proposed for the cancer patient, both as regards the diagnosis of first-detection neoplasms, and for the management of oncological emergencies and the palliation of symptoms.

Reviewer 2 Report
Dear Authors,
thank You for this comprehensive review of oncologic emergencies. I believe that the text may benefit from some editorial interventions.
1. Actually, You have not precised the exact aim of this paper in the abstract and introduction part. It would be better for the reader that the text is focused on clinical aspects of oncological emergencies.
2. In my opinion one oncologic emergency is missing which is anemia developing secondarily to neoplasmatic disease or therapy. Please, add a paragraph on that focusing on signs/symptoms and on indications for acute blood transfusions in oncologic patients.
3. I feel like your advices upon diagnosing and therapy of the emergencies are too general. I would suggest adding more data on exact laboratory cut-off points and dosage of drugs.
4. As part of the readers may be emergency physicians I suggest adding graphs - algorithms - on solving the emergencies.
Author Response
- Actually, You have not precised the exact aim of this paper in the abstract and introduction part. It would be better for the reader that the text is focused on clinical aspects of oncological emergencies.
Dear reviewer,
thank you for your clarification and for your request.
Thanks for the suggestion, we thought we can underline this aspect in the modified abstract above, in the introduction, and in materials and methods:
Introduction. Starting from this assumption, knowledge of the main oncological emergencies is essential both from a clinical and treatment point of view, in order to understand and underline their impact in the field of emergency medicine. Cancer emergencies represent a health problem that is too little discussed and that should be adequately understood even by the emergency medicine physician. In particular, this represents a fundamentally important topic, due to the increase in cancer diagnoses and therefore the number of patients who can potentially manifest them. Their importance derives above all from the fact that, as is known, rapid diagnosis and timely treatment are able to determine an improvement in the prognosis within this group of patients. The emergency unit proves to be a fundamental and central unit in the management of the cancer patient. Many cancer cases are diagnosed in the first instance as a result of symptoms that lead the patient to the emergency room.
Materials and methods. To write this narrative review, with the aim to analize and give to the emergency medicine physician a focused and useful knowledge of cancer emergencies, a search was conducted over the period January 1981 - April 2023 using the main scientific platforms, including Pubmed, Scopus, Medline, Embase and Google scholar, analyzing 156 papers.
- In my opinion one oncologic emergency is missing which is anemia developing secondarily to neoplasmatic disease or therapy. Please, add a paragraph on that focusing on signs/symptoms and on indications for acute blood transfusions in oncologic patients.
Dear reviewer,
thank you for your clarification and for your request.
Thanks for the suggestion, we thought we’d write a paragraph about anemia, as follows:
“3.6 Anemia
Anemia is one of the most common laboratory abnormalities and represents one of the most common diagnoses in cancer patients. In this subgroup of patients, the causes of anemia are often multifactorial, related both to direct and indirect effects of the neoplasm, and to the effects of pharmacological treatments (mainly chemotherapy). [157-158]
By anemia, as is known, we mean a condition in which the concentration of hemoglobin (Hb) and/or the number of red blood cells are lower than the normal limit, and therefore insufficient to satisfy the physiological needs of an individual, above all in terms of of oxygen transport. [159-160]
According with World Health Organization (WHO) criteria and the CTCAE (v5.0) grading system for anemia, the grading of anemia in cancer patient is reported in the Table below.
|
Anemia grade |
Women [Hb] |
Men [Hb] |
|
Grade 1 |
Hb 11.9-10.0 g/dL |
Hb 12.9-10.0 g/dL |
|
Grade 2 |
Hb 9.9-8.0 g/dL |
Hb 9.9-8.0 g/dL |
|
Grade 3 |
Hb ≤7.9 g/dL |
Hb <7.9 g/dL |
|
Grade 4 |
Life-threatening consequences requiring urgent intervention, such as RBC transfusion |
Life-threatening consequences requiring urgent intervention, such as RBC transfusion |
Cancer-related anemia is due to three main mechanisms:- ineffective erythropoiesis- hemolysis- loss of blood.These mechanisms can lead to anemia individually, or in combination.
However, the causes of anemia can best be subclassified into three broad, related categories: production, destruction, and loss (bleeding).
These causes can then be related to three categories:
- drug induced
- induced by infectious diseases
- induced by the tumor itself, both solid tumors (especially gastrointestinal tumors) and blood tumors (acute or cronic leukemia)
- induced by vitamin deficiencies (vitamins B9, B12). [158; 161-163]
The table below shows the mechanisms of anemia induced by the different categories of oncological drugs.
|
Production (decreased) |
Destruction (increased) |
Loss (overt, occult, or iatrogenic bleeding) |
|
Myelosuppressive chemotherapy |
Antibiotics (eg, β-lactams, dapsone) |
Anticoagulation (eg, DOACs, LMWHs, warfarin) |
|
Radiation therapy |
Chemotherapy (eg, gemcitabine) |
Antiplatelet agents (eg, clopidogrel, prasugrel, ticagrelor) |
|
Tyrosine kinase inhibitors (delayed maturation) |
Immunotherapy (eg, nivolumab, pembrolizumab, ipilimumab) |
Nonsteroidal anti-inflammatory drugs |
|
Immunotherapy (inflammation) |
Intravenous immunoglobulin G |
Over-the-counter supplements (eg, turmeric) |
Therefore, with this in mind, a systematic approach is needed to identify the causes that lead to anemia. On the other hand, it must be remembered that a first detection of anemia must be carefully investigated, taking into consideration, among the possible differential diagnoses, also the presence of a solid or haematological neoplasm. Furthermore, the correct differential diagnosis of the cause of anemia in patients with cancer is essential to establish the correct treatment and reduce transfusion requirements, with the ultimate aim of improving both quality of life and survival, through improvement in cognition, fatigue, and exercise tolerance. [157-159; 164]
However, blood transfusions in cancer patients, used in the presence of risk for life but also in anemia of grades other than 4, depending on the needs and clinical status of the patient, have been linked to an increased risk of thrombosis, transmission of pathogens, transfusion reactions, volume and iron overload, and ultimately also decreased survival. In fact, red blood cell transfusions are associated with a 10-fold greater risk of morbidity than IV iron (1 in 21,413 for RBC vs ~1 in 200,000 for current IV iron products). 60 Furthermore, it is useful to underline that although each unit of PRBC contains about 250 mg of iron, this is not immediately bioavailable due to the average life of the transfused red blood cell (90 days). It is therefore the clinician's duty to consider both the clinical indicators and the laboratory analyzes in order to lean towards one or the other treatment, remembering that the transfusion of red blood cells, which requires about 1 or 2 hours per unit to be administered, remains an option for the treatment of anemia associated with grade 2 to 4 cancer when other therapies have failed. [165-168]
- Anand S, Burkenroad A, Glaspy J. Workup of anemia in cancer. Clin Adv Hematol Oncol. 2020 Oct;18(10):640-646. PMID: 33201870.
- Gilreath JA, Rodgers GM. How I treat cancer-associated anemia. Blood. 2020 Aug 13;136(7):801-813. doi: 10.1182/blood.2019004017. PMID: 32556170.
- Chaparro CM, Suchdev PS. Anemia epidemiology, pathophysiology, and etiology in low- and middle-income countries. Ann N Y Acad Sci. 2019 Aug;1450(1):15-31. doi: 10.1111/nyas.14092. Epub 2019 Apr 22. PMID: 31008520; PMCID: PMC6697587.
- World Health Organization. Haemoglobin concentrations for the diagnosis of anaemia and assessment of severity. No. WHO/NMH/NHD/MNM/11.1. World Health Organization, 2011.
- Macciò A, Madeddu C, Gramignano G, Mulas C, Tanca L, Cherchi MC, Floris C, Omoto I, Barracca A, Ganz T. The role of inflammation, iron, and nutritional status in cancer-related anemia: results of a large, prospective, observational study. Haematologica. 2015 Jan;100(1):124-32. doi: 10.3324/haematol.2014.112813. Epub 2014 Sep 19. PMID: 25239265; PMCID: PMC4281325.
- Steensma DP. Clinical Implications of Clonal Hematopoiesis. Mayo Clin Proc. 2018 Aug;93(8):1122-1130. doi: 10.1016/j.mayocp.2018.04.002. Epub 2018 Jul 4. PMID: 30078412.
- Spivak JL. Cancer-related anemia: its causes and characteristics. Semin Oncol. 1994 Apr;21(2 Suppl 3):3-8. PMID: 8202724.
- Crawford J, Cella D, Cleeland CS, Cremieux PY, Demetri GD, Sarokhan BJ, Slavin MB, Glaspy JA. Relationship between changes in hemoglobin level and quality of life during chemotherapy in anemic cancer patients receiving epoetin alfa therapy. Cancer. 2002 Aug 15;95(4):888-95. doi: 10.1002/cncr.10763. PMID: 12209734.
- Spivak JL, Gascón P, Ludwig H. Anemia management in oncology and hematology. Oncologist. 2009;14 Suppl 1:43-56. doi: 10.1634/theoncologist.2009-S1-43. PMID: 19762516.
- Khorana AA, Francis CW, Blumberg N, Culakova E, Refaai MA, Lyman GH. Blood transfusions, thrombosis, and mortality in hospitalized patients with cancer. Arch Intern Med. 2008 Nov 24;168(21):2377-81. doi: 10.1001/archinte.168.21.2377. PMID: 19029504; PMCID: PMC2775132.
- Iqbal N, Haider K, Sundaram V, Radosevic J, Burnouf T, Seghatchian J, Goubran H. Red blood cell transfusion and outcome in cancer. Transfus Apher Sci. 2017 Jun;56(3):287-290. doi: 10.1016/j.transci.2017.05.014. Epub 2017 May 26. PMID: 28602484.
- Lee J, Chin JH, Kim JI, Lee EH, Choi IC. Association between red blood cell transfusion and long-term mortality in patients with cancer of the esophagus after esophagectomy. Dis Esophagus. 2018 Feb 1;31(2). doi: 10.1093/dote/dox123. PMID: 29077842.
- I feel like your advices upon diagnosing and therapy of the emergencies are too general. I would suggest adding more data on exact laboratory cut-off points and dosage of drugs.
Dear reviewer,
thank you for your clarification and for your request.
Thanks for the suggestion, we thought we'd create tables according to your advice, as follows:
- Table including presentation and management of Neurological emergencies
Table 1. Presentation and Management of Neurological Emergencies: Spinal cord compression (scc) and intracranial hypertension.
|
Syndrome |
Presentation |
Management |
|
Spinal cord compression |
v A gradual onset may only be demonstrated by back pain and weakness
v Symptoms: acute sensory loss, urinary retention, constipation, perineal numbness, sudden inability to walk |
v Neurologic examination
v total spine MRI: gold standard. For patients unable to tolerate MRI, a CT scan with and without IV contrast can be attempted, but the diagnostic yield because of inferior imaging resolution is limited. v Opiate pain control and dexamethasone 10 mg IV loading dose followed by 4 mg every 6 h
v urgent neurosurgical and radiation oncology consultation
v If surgery is not indicated, palliative stereotactic body radiotherapy of 16-24 Gy in 1 fraction or 24-30 Gy in 3 fractions is administered |
|
Intracranial hypertension |
v Symptoms: intractable headaches or nausea and vomiting are most common; (vomiting is correlated with infratentorial lesions and risk brainstem impingement), altered mental status, focal neurologic deficits, vision changes |
v An expedited noncontrast CT scan to evaluate for midline shift is then followed by a nonurgent brain MRI with and without contrast to further characterize the posterior fossa and delineate smaller lesions
v Dexamethasone is recommended except when lymphoma is suspected because this can compromise the diagnostic biopsy by shrinking the mass v Nonenzyme-inducing anticonvulsants (eg, levetiracetam) are typically administered in oncology patients because the risk of drug-drug interactions is minimized, although there are mixed data supporting their use
v Hypertonic (3%) saline or mannitol can be used emergently to create an osmotic gradient across the blood-brain barrier, the priority is alleviating the mass effect
v Gamma Knife stereotactic radiosurgery has emerged as the preferred modality for managing solid intracranial masses
v Lumbar puncture is contraindicated, because increases the risk for herniation |
- Table including presentation and management about SIADH (Syndrome of inappropriate antidiuretic hormone) and Hypercalcemia
Table 2. Presentation and Management of Endocrine and Metabolic Emergencies
|
Syndrome |
Presentation |
Management |
|
Syndrome of inappropriate antidiuretic hormone (SIADH) |
v Urine osmolarity >100 mOsm (coincident with euvolemic hypotonic hyponatremia)
v Corresponding serum Na levels: - Mild: 130 – 134 mEq/dL - Moderate: 125 – 129 mEq/dL - Severe: <125 mEq/dL
v Acute hyponatremia with headaches or neurocognitive slowing
v Severe hyponatremia can be associated with seizures or death.
REMEMBER:
v Must be distinguished from hypovolemic hyponatremia (urine osmolarity >300 mOsm; urine sodium < 20 mEq/L). Main cause: excessive gadtrointestinal loss.
|
v Symptomatic hyponatremia: 100 mL 3% normal saline bolus to acutely raise serum sodium by 2-3 mEq/L.
v Chronic hyponatremia: Free water restriction and sodium chloride tablets. REMEMBER: v Total increase of serum sodium by no more than 4-6 mEq/L in 24 hrs to avoid central pontine myelinolysis.
v Correction of hyponatremia is usually necessary prior to initiating systemic therapy.
|
|
Hypercalcemia
|
v Corresponding serum Ca levels: - Mild: 10.5 – 11.9 mg/dL - Moderate: 12.0 – 13.9 mg/dL - Severe: ≥14.0 mg/dL
v If hypoalbuminemia, observed serum calcium must be further increased by 0.8*(4.0 – serum albumin) mg/dL.
v Presentation: altered mental status, muscle weakness, constipation, dehydration with ensuing acute kidney injury, urolithiasis (in subacute presentations).
|
v Immediate aggressive intravenous hydration with normal saline (1-2 L in the first hour, followed by 2 L at 200 mL/hr with close monitoring of volume status). v Early intravenous bisphosphonate administration (most commonly zolendronic acid). v Supplemental calcitonin can be administered during the first 48 hours while the bisphosphonate is not yet at peak efficacy. Denosumab is alternatively administered in bisphosphonate-refractory cases. v Loop diuretics are generally now avoided as they can exacerbate hypercalcemia and kidney injury in inadequately hydrated patients. (LeGrand 200826)
|
- Table including presentation and management of Vascular Emergencies
Table 3. Presentation and Management of Vascular Emergencies: Superior Vena Cava Syndrome and Venous Thromboembolism
|
Syndrome |
Presentation |
Management |
|
Superior Vena Cava Syndrome |
v Facial edema and subcutaneous vein engorgement in head, neck, and chest
v Complete obstructions additionally present with plethora, dyspnea, orthopnea, cough, hoarseness, cyanosis, headache, seizures, and eventually coma
v Chest X-ray: mediastinal widening (66%) or pleural effusions (25%)
v CT with contrast: gold standard.
|
v Elevate head to minimize venous congestion
v Medical and radiation oncology consultation should be expedited to initiate systemic therapy because reducing tumor bulk is definitive
v Urgent thrombolysis, thrombectomy, or placement of a venous stent may alleviate stridor and hemodynamic compromise although vascular intervention risks luminal perforation
v Diuretic use should be minimized
|
|
Venous Thromboembolism |
v Chief complaints of shortness of breath, unilateral leg swelling, or reduced oxygenation on pulse oximetry. v D-dimer levels are not informative: they can be elevated generally in cancer patients |
v CT angiography of the chest: is the definitive study because it not only can rule out other processes but also can confirm right ventricular strain
v If IV contrast is contraindicated, a ventilation-perfusion scan along with cardiac echography is appropriate
v Systemic thrombolysis is indicated for massive PE with hemodynamic compromise except in patients with a high risk of bleeding, for whom catheter-assisted thrombectomy is indicated
v Factor Xa inhibitors are noninferior to low-molecular-weight heparin with apixaban demonstrating fewer major bleeding events
v The benefit of thromboprophylaxis has not been demonstrated
v Patients with small, incidental PEs and no functional or vital sign compromise are eligible to initiate anticoagulation in the ED then be safely discharged home with close follow-up |
- Table including presentation and management of Febrile Neutropenia
Table 4. Presentation and Management of Febrile Neutropenia
|
Syndrome |
Presentation |
Management |
|
Superior Vena Cava Syndrome |
v A single oral temperature measured at ≥38.3 °C (101 °F) or a temperature of ≥38.0 °C (100.4 °F) sustained over a 1-h period combined with severe neutropenia, defined as an absolute neutrophil count (ANC) <500 cells/mm3 or an ANC that is expected to decrease to <500 cells/mm3 during the next 48 h
v Presentation can bridge the spectrum from otherwise asymptomatic to moribund with florid hemodynamic collapse.
|
v The current door-to-antibiotics guideline is <1 h after ED presentation, although recent research has shown that delays up to 3 h do not substantially affect outcomes
v Blood cultures, both peripheral and from in-dwelling lines, should be universally collected
v Empiric monotherapy with an antipseudomonal β-lactam is recommended with the following exceptions: - Carbapenems for patients with a high risk of infection by an extended-spectrum β-lactamase-expressing organism - Polymyxin-colistin in patients at high risk for a carbapenemase-producing Klebsiella - Aztreonam, fluoroquinolones, or aminoglycosides for patients with an anaphylactic β-lactam allergy
v Vancomycin, linezolid, or daptomycin should be given only when either methicillin-resistant Staphylococcus aureus or indwelling catheter infections are likely and should be discontinued once these are ruled out
v Empiric antifungal coverage is not recommended at the time of presentation |
- Table including presentation and management of Severe Anemia
Table 5. Presentation and Management of Severe Anemia
|
Syndrome |
Presentation |
Management |
|
Severe anemia |
v Symptoms: debilitating fatigue and dyspnea, initially with exertion then at rest |
v Laboratory exams: reticulocyte count and haptoglobin, iron studies, serum vitamin B12 and folate levels, liver function tests (including albumin, prealbumin, total and direct bilirubin), and lactate dehydrogenase concentrations as well as a peripheral blood smear should be obtained in addition to the complete blood count
v hemagglutinins can be assessed to rule out hemolysis
v Transfusion should be used as sparingly as possible and should target a hemoglobin level of 7.0 mg/dL unless the patient has established cardiac disease or continues to be overtly symptomatic where 8.0 mg/dL is appropriate
v Erythropoiesis-stimulating agents are no longer indicated for cancer-related anemia |
- As part of the readers may be emergency physicians I suggest adding graphs - algorithms - on solving the emergencies.
Dear reviewer,
thank you for your clarification and for your request.
We decided, for a more comprehensive work, to reply to this point together with the previous one.

Round 2
Reviewer 1 Report
Thank you for your reply. Unfortunately I can not see that you included the changes stated in your reply in the revised manuscript (probably a wrong file has been uploaded?). Please revise the manuscript as described in your reply.
Author Response
Dear Reviewer,
There is probably an error in the review mode. We have now attached a file that should remedy the problem.
Reviewer 2 Report
Dear Authors,
I am very thankful for your modification within the manuscript. In my opinion the paper is even more clinically useful.
I have no further comments or questions.
Author Response
Dear reviewer, thank you.
Round 3
Reviewer 1 Report
Dear authors,
thank you for the revision. I think the paper is now easier to read and understand.
English language proofreading is needed before the manuscript can be accepted for publication.
English language proofreading is needed before the manuscript can be accepted for publication.